# Structure of cryptophyte photosystem II–light-harvesting antennae supercomplex

Yu-Zhong Zhang [1,2,3,6], Kang Li [2,3,6], Bing-Yue Qin[1], Jian-Ping Guo[4], Quan-Bao Zhang[1], Dian-Li Zhao[3], Xiu-Lan Chen [1,3], Jun Gao [4], Lu-Ning Liu [2,5] & Long-Sheng Zhao[1,3]

Cryptophytes are ancestral photosynthetic organisms evolved from red algae through secondary endosymbiosis. They have developed alloxanthin-chlorophyll *a/c2*-binding proteins (ACPs) as light-harvesting complexes (LHCs). The distinctive properties of cryptophytes contribute to efficient oxygenic photosynthesis and underscore the evolutionary relationships of red-lineage plastids. Here we present the cryo-electron microscopy structure of the Photosystem II (PSII)–ACPII supercomplex from the cryptophyte *Chroomonas placoidea*. The structure includes a PSII dimer and twelve ACPII monomers forming four linear trimers. These trimers structurally resemble red algae LHCs and cryptophyte ACPI trimers that associate with Photosystem I (PSI), suggesting their close evolutionary links. We also determine a Chl *a*-binding subunit, Psb-γ, essential for stabilizing PSII–ACPII association. Furthermore, computational calculation provides insights into the excitation energy transfer pathways. Our study lays a solid structural foundation for understanding the light-energy capture and transfer in cryptophyte PSII–ACPII, evolutionary variations in PSII–LHCII, and the origin of red-lineage LHCIIs.

Oxygenic photosynthesis is one of the most vital processes for converting sunlight into chemical energy and supporting all life on Earth[1–3]. An essential protein complex in oxygenic photosynthesis is Photosystem II (PSII), a multi-subunit pigment–protein complex embedded in thylakoid membranes. PSII is instrumental in catalyzing light-induced water-splitting reactions, which result in the generation of electrons, protons, and molecular oxygen. The core of PSII is associated with peripheral light-harvesting antennae to enhance the absorption cross-section of PSII. Solar light is captured by antennae, and is then transferred to PSII, triggering the primary charge separation. In cyanobacteria and red algae, the antenna systems of PSII are the soluble phycobilisomes[2,4–11]. In contrast, green algae and higher plants employ membrane-integrated LHCII[12], which bind with numerous pigments, including chlorophylls (Chls) and carotenoids, forming the PSII–LHCII supercomplex.

Among the PSII–LHCII supercomplexes of oxygenic photosynthetic organisms, the PSII core subunits are highly conserved, whereas the peripheral light-harvesting antenna are diversified. The number and organization of LHCII are distinct among eukaryotic organisms. For instance, green algae and higher plants typically form LHCII trimers in addition to monomers[13–18]. Diatoms, on the other hand, form not only LHCII monomers but also dimers, trimers, and tetramers[19–24]. This structural diversity allows for more LHCII units to associate with the PSII core within a confined space.

[1]Marine Biotechnology Research Center, State Key Laboratory of Microbial Technology, Shandong University, Qingdao, China. [2]MOE Key Laboratory of Evolution and Marine Biodiversity, Frontiers Science Center for Deep Ocean Multispheres and Earth System & College of Marine Life Sciences, Ocean University of China, Qingdao, China. [3]Laboratory for Marine Biology and Biotechnology, Qingdao Marine Science and Technology Center, Qingdao, China. [4]Hubei Key Laboratory of Agricultural Bioinformatics, College of Informatics, Huazhong Agricultural University, Wuhan, China. [5]Institute of Systems, Molecular and Integrative Biology, University of Liverpool, Liverpool, UK. [6]These authors contributed equally: Yu-Zhong Zhang, Kang Li. ✉e-mail: zhangyz@sdu.edu.cn; gaojun@mail.hzau.edu.cn; luning.liu@liverpool.ac.uk; zhaols@sdu.edu.cn

Cryptophytes are key members of the red lineage and serve as primary producers in a wide range of ecological habitats[25–29]. The membrane-embedded LHCs of cryptophytes contain Chl $a/c_2$ and alloxanthins, thereby designated alloxanthin Chl $a/c$-binding proteins (ACPs). The structure of cryptophytes Photosystem I-ACPI (PSI−ACPI) highlights the evolutionary link between red algae, cryptophytes, and diatoms, which emerged during secondary endosymbiosis[30]. Recent studies have revealed the detailed structures of the PSII−LHCII complexes in red algae (PSII−phycobilisome)[7,31] and from diatoms (PSII−FCPII, which contains fucoxanthin-chlorophyll $a/c$-binding proteins)[19–24]. However, the high-resolution structure of the PSII−ACPII complex in cryptophytes has not yet been characterized.

Here, we report the structure of a PSII−ACPII supercomplex from the cryptophyte *Chroomonas placoidea* by cryo-electron microscopy (cryo-EM). The structure reveals that PSII−ACPII consists of a PSII dimer and twelve ACPII monomers forming four linear trimers, as well as a Chl $a$-binding Psb-γ subunit that is identified to connect ACPIIs with the PSII core. Our study provides insight into the distinct assembly patterns of LHCII with the PSII core and the evolutionary diversity of PSII−LHCII supercomplexes in the red lineage. Furthermore, determination of the precise location and arrangement of pigments allows the mapping of light harvesting and energy transfer pathways in the cryptophyte PSII−ACPII.

## Results and discussion
### Overall structure of the PSII−ACPII supercomplex
The complete PSII−ACPII supercomplex was isolated from *C. placoidea* (Fig. S1, Supplementary Data 1), and its activity was confirmed by oxygen-evolution measurements ($128 \pm 13$ μmol $O_2$ (mg Chl)$^{-1}$ h$^{-1}$, $n = 3$), which was comparable to the activities of reported PSII

complexes[13,16,19,32,33]. The structure of PSII−ACPII was resolved by cryo-EM single-particle analysis, achieving an overall resolution of 2.53 Å (Fig. S2; Table S1). The cryptophyte PSII−ACPII supercomplex exhibited dimeric features, with a dimension of approximately 245 × 190 × 115 Å$^3$ and a molecular mass of about 1.1 MDa (Fig. 1). Each PSII monomer comprises a PSII core complex, 6 ACPIIs, and a Chl $a$-binding polypeptide, denoted as Psb-γ. Intriguingly, the ACPII antennae are exclusively monomers located along the shorter sides of the PSII core (Fig. 1a−c), distinct from the trimeric LHCII assemblies found in higher plants and green algae[13–18], or the dimeric and tetrameric configurations observed in diatom LHCII[19–24]. We also identified 209 chlorophylls $a$ (Chl $a$), 14 chlorophylls $c$ (Chl $c$), 4 pheophytin, 48 alloxanthin (Alx), 24 α-carotene (α-Car), 8 crocoxanthin (Cro), along with 4 plastoquinones, 4 hemes, 2 Mn$_4$CaO$_5$ clusters, 2 nonheme irons, 2 bicarbonate ions, 2 bound chloride ions, as well as 56 lipid molecules (including 16 distearoylmonogalactosyl diglyceride (LMG), 24 dipalmitoylphosphatidyl glycerol (LHG), 8 digalactosyldiacyl glycerol (DGDG), and 8 sulfoquinovosyldiacyl glycerol (SQDG)) (Fig. S3; Table S3). These findings corroborate the pigment analysis performed using high-performance liquid chromatography (HPLC) (Fig. S1).

### Structure of the PSII core
The PSII core is composed of 22 subunits (D1, CP47, CP43, D2, PsbE-F, PsbH-M, Psb30, PsbT, PsbW, PsbX, PsbZ), 4 extrinsic subunits (PsbQ', PsbO, PsbU, PsbV), and one Psb-γ polypeptide (Figs. 1, 2 and Fig. S3; Table S2). Notably, the cryptophyte PSII lacks PsbG, which is unique in diatom PSII and connects to the tetrameric LHCII, the PsbY subunit specific to red algae, as well as the Psb31 and Psb34 subunits found in both red algae and diatoms (Fig. 1 and Fig. S4; Table S2). All the core subunits are highly conserved, except for Psb-γ, which is absent in

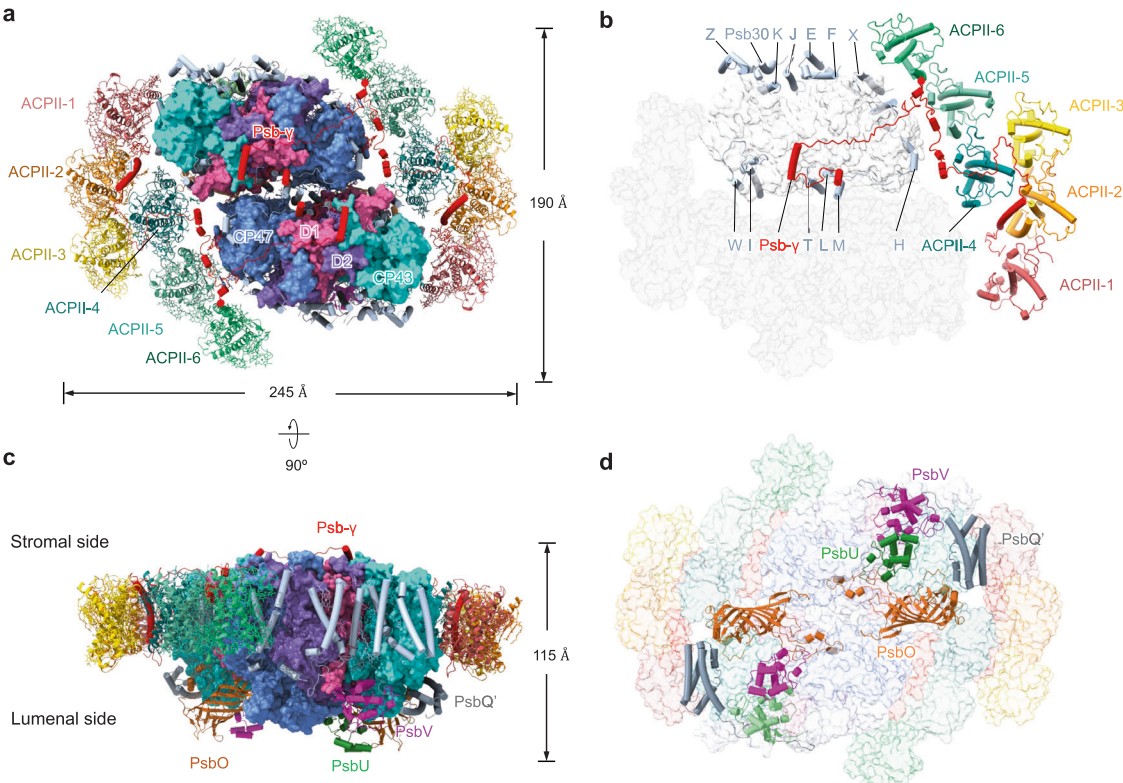

**Fig. 1 | Overall structure of the cryptophyte PSII−ACPII supercomplex.**
**a** PSII−ACPII supercomplex viewed from the stromal side. Four main core subunits are labeled by their names. LHCs are labeled as ACPII. **b** Structure of the monomer of PSII−ACPII supercomplex viewed from the same direction as in (**a**). Small subunits of the PSII core were indicated with one letter. **c** Side view of the PSII−ACPII supercomplex with the four extrinsic proteins labeled, colors of the subunits are the same as in (**a**). **d** PSII−ACPII supercomplex viewed from the lumenal side with the four extrinsic proteins labeled.

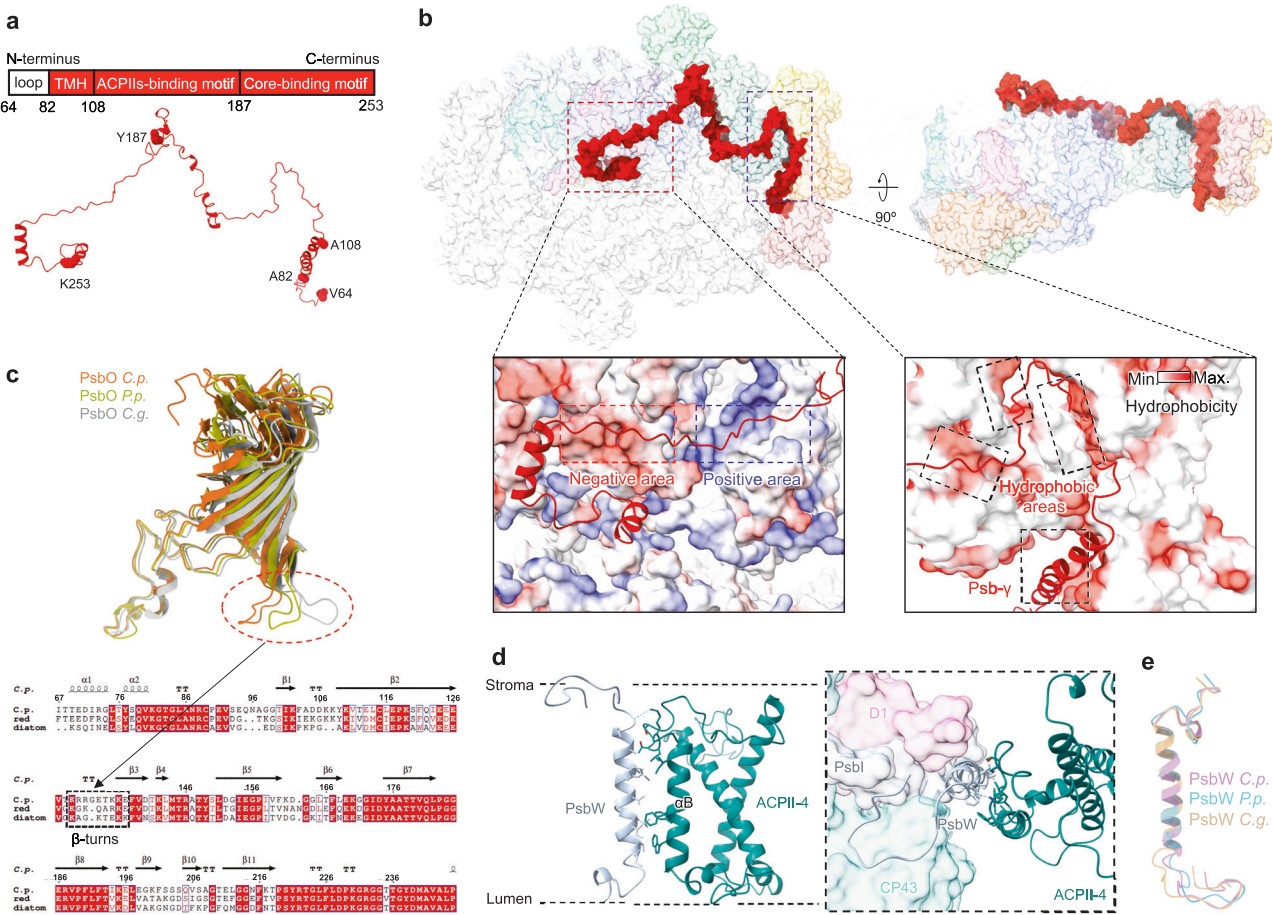

**Fig. 2 | Locations and structures of the subunits Psb-γ, PsbO, and PsbW.**
**a** Representative diagrams of the structural elements of Psb-γ are shown above the structure. **b** Overview of Psb-γ interacting with PSII–ACPII supercomplex viewed from the stromal side. Electrostatic surface representation of the PSII core (red dash box) is shown in the left panel. Hydrophobic interactions between Psb-γ and ACPIIs (purple dash box) are shown in the right panel. **c** Structural comparison of PsbO from *Cryptophyte* (orange) with red algae (yellow, PDB: 7Y5E) and diatom

(light gray, PDB: 6LY5) with their sequence alignment on the right panel. The loop with different positions was indicated with dash lines. **d** The location and interactions of PsbW with ACPII-4. **e** Structural comparison of PsbW from cryptophyte *Chroomonas placoidea* (*C.p.*) with red alga *Porphyridium purpureum* (*P.p.*) (PDB: 7Y5E) and diatom *Chaetoceros gracilis* (*C.g.*) (PDB: 6LY5). Min., minimum; max., maximum.

other PSII–LHCII supercomplexes previously characterized. The locations and structures of $Q_A$ and the head group of $Q_B$ are highly conserved among different PSII–LHCIIs (Fig. S4c). By contrast, the tail of $Q_B$ exhibits diverse conformations and orientations, suggesting the conformational flexibility of $Q_B$ compared to $Q_A$ in different PSII–LHCII structures. The arrangement of the oxygen-evolving complex (OEC) remains consistent across various PSII–LHCII complexes, with the majority of water molecules surrounding the OEC showing similar patterns (Fig. S4d), highlighting the conservation of the PSII core structure.

A distinctive feature of the cryptophyte PSII core is the presence of Psb-γ (PDB: 8XR6; Chain N, n), a linker protein that acts as a molecular thread to "stitch" subunits in the supercomplex. From the N-terminus to the C-terminus, Psb-γ encompasses the N-terminal region, the transmembrane domain, the ACPIIs-binding motif and the core-binding motif, establishing a connection between the entire core and ACPII (Fig. 2a, b). Specifically, the N-terminal loop of Psb-γ attaches to the lumenal surface, swings towards the PsbQ'–ACPII-1 side, and binds to the α-helix of PsbQ' and the N-terminus of ACPII-1 through electrostatic interactions (Fig. 2a and Fig. S5a, b). Psb-γ has a single transmembrane (TM) α-helix (A82-A108) that extends into the interface between the core, ACPII-2, and ACPII-4 (Fig. S5a, c). The ACPII-binding motif of Psb-γ, which spans the stromal side of PSII, interacts with ACPII-2/3/4/5/6, D1, CP47,

and PsbH predominantly through electrostatic interactions (Fig. 2a, b). In addition, there are grooves formed by ACPIIs and a space between them. These grooves form platforms to stabilize Psb-γ and ACPIIs through hydrophobic interactions (Fig. 2b). Intense hydrogen bonds and salt bridges were formed among Psb-γ, ACPIIs, and the PSII core subunits (Fig. S5b–f), suggesting their strong connections. It is noteworthy that Y150/Psb-γ forms π–π interactions with Y74/ACPII-4 (Fig. S5e), while Y200/Psb-γ and F234/Psb-γ form two cation·π interactions with R127/CP47 and K31/PsbT, respectively (Fig. S5f). The core-binding motif and the c-terminal region of Psb-γ may play a role in the assembly, stabilization, and dimerization of the PSII core due to their interactions.

Sequence alignment reveals that there is only one Psb-γ homolog found in the cryptophyte *Guillardia theta* (G.t.) (Fig. S6a), and no homologous sequences are identified in red algae or diatoms. Intriguingly, Psb-γ and two linker proteins of red algae, $L_{PP}1$ and $L_{RC}3$, share structural features at the core-binding domain and have a similar binding position to the PSII core (Fig. S7). However, Psb-γ lacks the membrane-extrinsic domains found in $L_{PP}1$ and $L_{RC}3$, which are essential to stabilize the coupling of phycobilisomes and PSII at the stromal surface[7]. Cryptophytes do not possess phycobilisomes; instead, they produce phycobiliproteins that are specifically localized inside the thylakoid lumen for capturing light and transferring energy to photosystems[34,35]. The unoccupied area on the stromal surface of

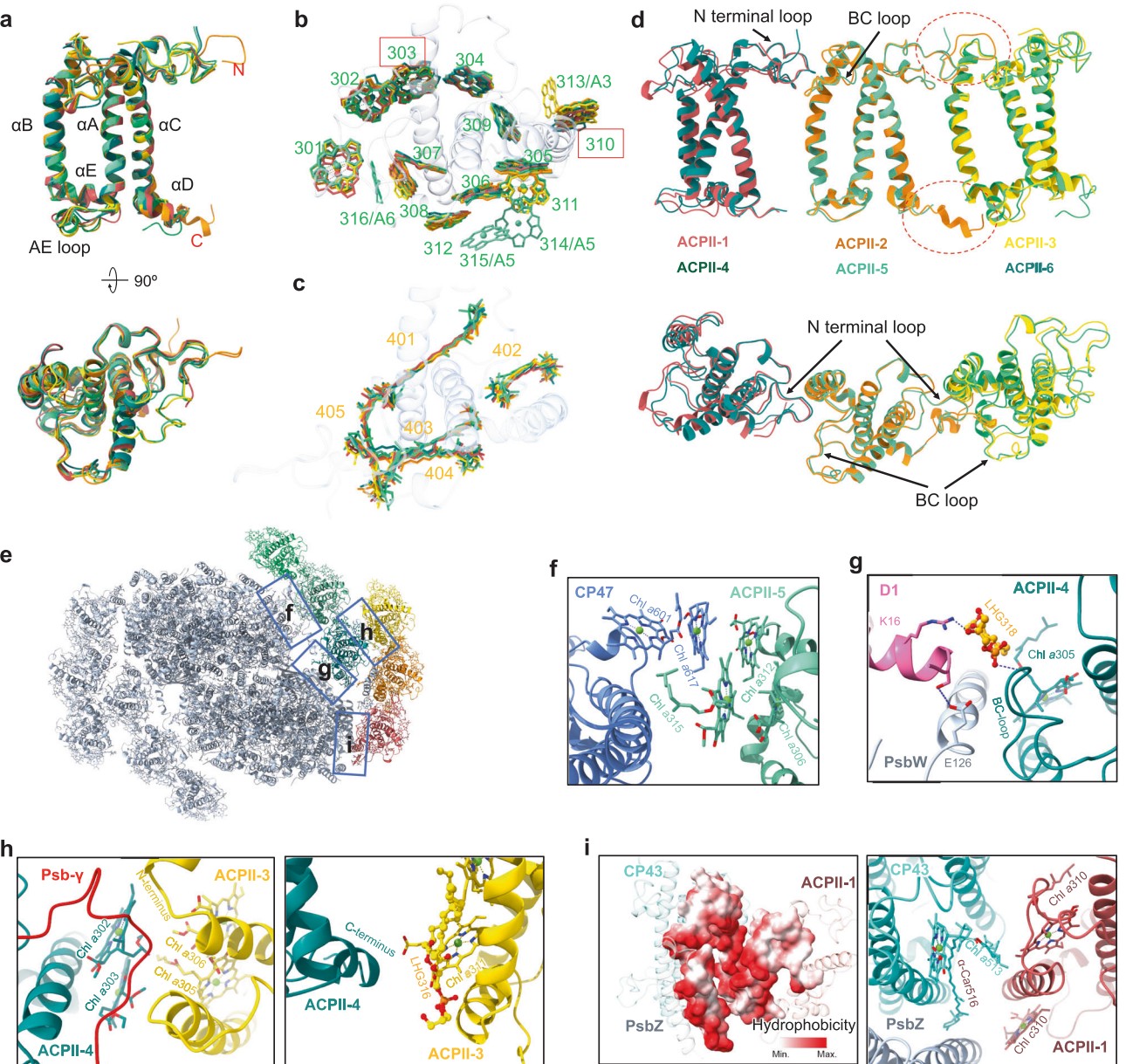

**Fig. 3 | Structures and pigment-binding sites of ACPIIs and important interactions between them and PSII core. a** Structural comparison of 6 ACPIIs. **b** Superposition of the Chl-binding sites of all ACPIIs viewed from the stromal side. Sixteen Chl sites are identified. The sites that could bind Chl *c* are indicated by red boxes. **c** Superposition of the carotenoid sites of all ACPIIs viewed from the stromal side. Five carotenoid sites are identified. Colors for the ACPIIs and their pigments in (**b**), (**c**) are in line with those in Fig. 1. **d** Superposition of ACPIIs in three adjacent modes. Circled areas with red dash lines indicate long loops of ACPII-2. **e** Overall structure of the PSII–FCPII supercomplex. The areas circled by rectangles

correspond to the enlarged panels shown in the following, among which the interactions are described in (**f**), (**g**), (**h**), (**i**), respectively. **f** Interactions between ACPII-5 and CP47, at the stromal sides. **g** Interactions between ACPII-4, PsbW, and D1. **h** Interactions between ACPII-3 and ACPII-4, at the stromal and lumenal sides, respectively. **i** Interactions between ACPII-1, PsbZ, and CP43. Hydrophobic interactions between the ACPII-1 and core are shown in the left panel. Lipid molecules are shown as balls and sticks. The dashed lines in blue indicate hydrogen bonds between adjacent subunits.

PSII enables Psb-γ to bind to PSII, which is important for stabilizing the PSII–ACPII association (Fig. 2 and Fig. S5).

PsbO (Chain O, o) exhibits a high sequence identity and structural conservation across cryptophytes, red algae, and diatoms. However, there are variations in the location and orientation of the β-turns (K129-E137), which pivot towards CP47 of the adjacent core (Fig. 2c). PsbW (Chain W, w) is a peripheral TM helix (TMH) of PSII and is located near D1, PsbI, CP43 of the PSII core and the αB helix of ACPII-4 (Figs. 1 and 2d). Its homologous sequences were found in red algae but not in diatoms (Fig. S6b), and the homologous sequence in the red alga *Porphyridium purpureum* fits well with the density map of its PsbW

subunit whose sequence has not yet been determined in the PSII structure[7]. The sequence of PsbW identified here shows low sequence identity with PsbW in diatom PSII–FCPII and the homologous sequences of red algae and cryptophyte found in the National Center of Biotechnology Information termed as 'PsbW' (Fig. S6c). These findings suggest that the PsbW proteins in cryptophytes and diatoms may have different origins from those in red algae. Although the PsbW subunits of red algal PSII and cryptophyte PSII–ACPII have a low sequence identity with that of diatom PSII–FCPII, there is significant structural conservation between them (Fig. 2e). Moreover, the PsbW subunit of green algal PSII–LHCII[15,16] has a low sequence identity with

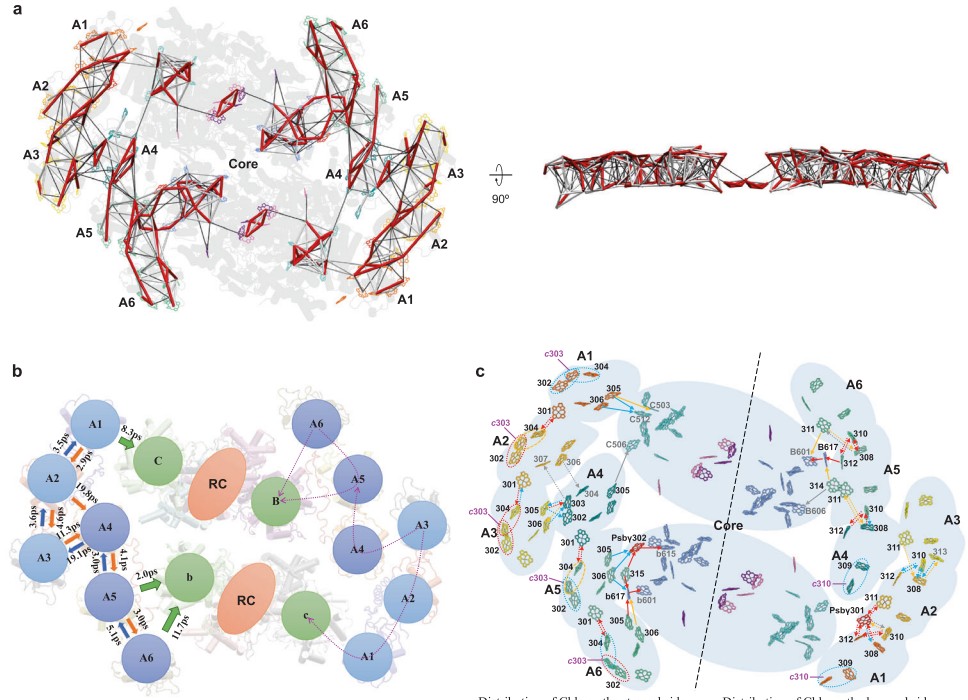

**Fig. 4 | Excitation energy transfer (EET) pathways in cryptophyte PSII–ACPII.** **a** Top view (left panel) and side view (right panel) of the inter-pigment EET time constant map. Rates faster than 1 ps (red line), in the 1–10 ps range (white line) and in the 10–20 ps range (black line). Inter-pigment rates slower than 20 ps are omitted. **b** EET time constant between ACPIIs and from ACPIIs to PSII core. The efficient EET pathways from ACPIIs to the PSII core are indicated by magenta arrows. **c** Chl distribution and EET pathways in PSII–ACPII viewed from the stromal side. Chls *c* were indicated magenta. Solid arrows: EET pathways from ACPIIs to PSII core; dashed arrows: EET pathways between the ACPIIs; dashed ovals: EET pathways between Chl *c* and surrounding Chls. red arrows and ovals: faster than 2 ps; blue arrows and ovals: between 2 ps and 5 ps; orange arrows and ovals: between 5 ps and 10 ps; gray arrows: between 10 ps and 20 ps.

those of red algal PSII, cryptophyte PSII–ACPII, and diatom PSII–FCPII (Fig. S6d). Phylogenic analysis revealed that green algal PsbW exhibits a distant evolutionary relationship with PsbW of red algae, cryptophytes, and diatoms (Fig. S6e). The residues involved in interactions are mostly conserved as charged or hydrophobic amino acids, signifying their vital role in stabilizing and connecting the components (Fig. 2d). In contrast to the diatom PsbW, which is relatively distant from the peripheral antenna[20], the cryptophyte PsbW is close to ACPII-4 and is assumed to develop more intricate interactions with ACPII-4, suggesting its potential role in stabilizing ACPIIs (Fig. 2d).

## ACPII structure

The cryptophyte PSII–ACPII supercomplex contains 12 ACPIIs, all of which are strictly monomeric. All ACPII apoproteins consist of three TMHs (αA, αB, αC) and a short amphipathic helix, αE, which is positioned between αA and αB, consistent with the cryptophyte APCIs that associate with PSI (Fig. 3a and Figs. S8–S10)[30]. Our structure shows that the 12 ACPIIs contain 133 Chl *a*, 14 Chl *c*, 48 Alx, 8 Cro, and 4 α-Car molecules (Table S3). The Chl *a/c* ratio of 9.5 is higher than those for cryptophyte ACPIs (7.95) and diatom FCPIIs (2.01); whereas the Chl/Car ratio (2.45) is comparable to that for cryptophyte ACPIs (2.39), both greater than that for diatom FCPIIs (1.75)[21,30] (Table S3). The amount of Chl *c* in FCPIIs is five times the level of Chl *c* in ACPIIs, and the amount of Car in FCPIIs is two times the level of Car in ACPIIs. The lower content of Chl *c* and Car molecules in ACPIIs leads to a lower blue-green light absorption capacity than FCPIIs (Fig. S1b). The large amount of Chl *c* in FCPIIs leads to the blue shift of diatom PSII–FCPII in the red absorption spectra compared with PSII–ACPII. The pigments Chl *c* and Car enable cryptophyte PSII–ACPII absorbs light in the blue-green region which could not be effectively absorbed by Chl *a*. The light absorption features facilitate the survival of cryptophytes in deep water, where the blue-green light can penetrate. Chls *c* transfer energy

efficiently to the coupled Chls *a*[36], and were proposed to facilitate energy transfer from Cars to Chls *a*[37]. In addition, Chl *c* has been hypothesized to play a role in energy dissipation under high-light conditions[38]. Thus, Chls *c* and Cars may form an energy-quenching system to protect PSII–ACPII from excess irradiation.

The Chl-binding sites are largely conserved among all ACPIIs (Fig. 3b; Tables S3, S4), except that ACPII-1/4 lack the Chl 311 site, ACPII-3 has one addition site (313), ACPII-6 has one addition site (316) and ACPII-5 has two additional binding sites (314 and 315). The Chls 314/315 are close to the PSII core, and the shortest distances of Chl 314 and Chl 315 to the PSII core pigment Chl $617_{PsbB}$ are 4.1 nm and 6.3 nm, respectively. This suggests the potential role of Chls 314/315 in protein stabilization and energy transfer between ACPII and the PSII core. Chl-binding sites 303 and 310 represent shared sites for either Chl *a* or Chl *c* (Fig. 3b). Moreover, each ACPII contains 5 conserved carotenoid-binding sites (401–405) (Fig. 3c; Table S4). Compared to green algal LHCIIs, cryptophyte ACPIIs have a similar number of pigment-binding sites, but the positions of some of these sites vary. Specifically, green algal LHCIIs have 14 conserved Chl-binding sites, whereas cryptophyte ACPIIs share the same positions with 9 of these (Fig. S9c). Moreover, green algal LHCIIs feature 4 conserved carotenoid-binding sites, whereas cryptophyte ACPIIs share 2 of them (sites 401, 403) (Fig. S9d).

In addition to the association between ACPIIs and Psb-γ, the interactions between adjacent ACPIIs within the same antenna layer are mainly formed between the BC loop of ACPII-(*n* + 1) and the N-terminal loop of ACPII-(*n*) as well as the pigments (Fig. 3d–i). In ACPII-2, the loops of the N- and C-termini are significantly longer compared to those in ACPII-5 or ACPI-13 at the corresponding positions, indicating a more robust interaction between ACPII-2 and ACPII-3 (Fig. 3d). The interactions between the outer and inner ACPIIs are mainly mediated by Psb-γ and the pigments between them (Fig. 3h).

Moreover, there are two lipid molecules in the periphery of ACPII-3/4, which enhance its interactions with ACPIIs and the PSII core (Fig. 3g, h).

The arrangements of LHCII in diatoms, green algae, and land plants are intricate, involving LHCII dimers, triangular trimers, and tetramers attaching to the PSII core through the mediation of LHCII monomers and specific subunits (PsbG and Psb34 in diatoms) (Fig. S11)[13–24]. By contrast, the arrangement of ACPIIs in cryptophyte PSII−ACPII is unique, as it lacks the LHCII dimers and tetramers found in diatoms and triangular trimers identified in green algae and land plants. ACPII-1/2/3 and ACPII-4/5/6 antennae form two belts of linear trimers on one side of the PSII core, and the ACPII-1/2/3 assemblies are positioned on the outer side of ACPII-4/5/6 (Fig. 1). Intriguingly, 3D variability analysis revealed that the peripheral ACPII subunits exhibit a high degree of orientational flexibility, highlighting the dynamic architecture of the PSII−ACPII supercomplex (Fig. S2). Similar structural flexibility has also been observed in other photosynthetic reaction center−light-harvesting antenna systems[39]. Compared to diatom FCPIIs, cryptophyte ACPIIs have a closer resemblance to red algal LHCRs in terms of sequence and structure, which explains the absence of the dimers or tetramers of ACPIIs in cryptophytes (Figs. S9b, S10).

Phylogenetic analysis revealed that all the ACPIIs belong to the Lhcr family, exhibiting a close genetic relationship with red algal LHCRs and cryptophyte APCIs (Fig. S12). In addition, ACPII-1 and ACPII-5 share the same protein sequences as ACPI-14 and ACPI-2, respectively. Overall, ACPII-1/2/3 and ACPII-4/5/6 form LHCII trimers with a strong conformational resemblance to the ACPI-14/13/12 and ACPI-3/2/1 trimers in cryptophyte PSI−ACPI as well as the Lhcr-5/4/3 trimer in red algal PSI−LHCR (Fig. S9)[7,30]. Furthermore, cryptophyte PSII and PSI share similar strategies for constructing their LHC network by using linear trimeric LHC modules assisted by Psb-γ and ACPI-S, respectively (Fig. S11d)[30]. Intriguingly, the overall structure composed of ACPII-1/2/3, ACPII-4/5/6, and TMH of Psb-γ is consistent with that composed of ACPI-14/13/12, ACPI-3/2/1, and TMH of ACPI-S in the cryptophyte PSI−ACPI (Fig. S11d). Our findings suggest that ACPIIs, as the initial membrane-integrated LHCIIs in red lineage PSII−LHCII, may evolve from ACPIs, both originating from red algal LHCRs. The linear trimeric structures may function as LHC building modules in the initial development of LHCs during the evolution of red lineage PSII−LHCII.

## Possible energy transfer pathways within the PSII−ACPII supercomplex

Using our structural data, we computationally simulated the excitation energy transfer (EET) between pigments within PSII−ACPII following the process as reported previously[40]. We assumed energy transfer in the limit of Förster theory, which is a generally accepted method for estimating the rate of energy transfer between two pigments that possess small interactions[41]. The EET rates (time constants) between different ACPIIs as well as between ACPIIs and the PSII core were estimated according to the generalized Förster theory, an extension of the classical Förster theory[42], to reflect the dynamics of the EET processes within PSII−ACPII.

The EET between Chls within PSII−ACPII occurs on the picosecond time scale through a dense network of pigments as shown in the EET time constant map (Fig. 4a, b), indicating rapid energy transfer within PSII−ACPII. The EET between ACPIIs is facilitated mainly by Chls 301/304 on the stromal side and Chls 308/310/311/312 on the lumenal side (Fig. 4c). In addition, the Chl 301$_{Psb-γ}$ mediates efficient EET on the lumenal side between ACPII-1 and ACPII-2. Energy transfer between ACPII-3 and ACPII-4 is mediated by the Chl 305/306 pair of ACPII-3 and Chl 302/303 pair of ACPII-4, forming an efficient EET pathway on the stromal side (Fig. 4c).

The EET from ACPIIs to the PSII core predominantly occurs through the mediation of ACPII-1/4/5/6, which are directly associated with the PSII core (Fig. 4b). Direct energy migration from ACPII-2 and ACPII-4 to the PSII core is less efficient, given their greater pigment−pigment distances. Instead, their energy can be transferred efficiently to ACPII-1 and ACPII-5, respectively, and then to the PSII core (Fig. 4b). ACPII-3 utilizes two distinct EET pathways (A3 → A2 → A1→core and A3 → A4 → A5→core) to the PSII core with similar rates (Fig. 4b).

On the stromal side, the EET from ACPII-1/4/5/6 to the PSII core is predominantly through the Chl 305/306 pair. In addition, the Chl 315$_{ACPII-5}$, Chl 302$_{Psb-γ}$, and Chl 617$_{PsbB}$ play critical roles in mediating EET from ACPII-5/6 to the PSII core (Fig. 4c). In ACPII-5, the Chl 305/306 pair and Chl 315$_{ACPII-5}$ transfer energy to the PSII core through Chl 302$_{Psb-γ}$ and Chl 617$_{PsbB}$ at a high rate (≤5 ps) (Fig. 4c). In ACPII-6, the Chl 305/306 pair forms an efficient EET route with Chl 617$_{PsbB}$. On the lumenal side, energy transfer from ACPII-5 to the PSII core is predominantly mediated by Chl 312$_{ACPII-5}$ and Chl 617$_{PsbB}$ (Fig. 4c). The Chl 314$_{ACPII-5}$ also forms EET with Chl 617$_{PsbB}$. Compared with the pigment arrangements in PSII−FCPII of diatom *Chaetoceros gracilis* and *Cyclotella meneghiniana*, the Chl 302$_{Psb-γ}$ and Chl 617$_{PsbB}$ in PSII−ACPII shorten the distance between Chls (Fig. S13c), implying faster EET from ACPII-5 to the core (Figs. S13a–c, S14a–c)[20,22]. Energy transfer from ACPII-6 to the PSII core on the lumenal side is mediated by Chl 311 at a rate of 9.6 ps, whereas the lumenal EET between ACPII-1 and the PSII core is less efficient (>20 ps) (Fig. 4c). Compared with ACPII-1/6, FCPIIs that are located at similar positions in PSII−FCPII of *C. gracilis* and *C. meneghiniana* adopt alternative EET pathways to the PSII core, and Chls in PsbZ which are absent in PSII−ACPII play a critical role in energy transfer (Figs. S13d and S14d, e). Chls of ACPII-4 are closer to the PSII core than those of Sm1 in *C. gracilis* and FCPII-H2 in *C. meneghiniana* and transfer energy directly to Chl 506$_{PsbC}$, whereas Sm1 transfers energy to Chl 506$_{PsbC}$ through Chl 101$_{PsbW}$ and Chl 102$_{PsbW}$ which are absent in PSII−ACPII (Figs. S13e and S14f). In addition, Chls of m1 in *C. gracilis* and FCPII-I in *C. meneghiniana* are situated in the same position as the space between ACPII-1 and ACPII-4, thereby mediating the EET from peripheral FCPIIs to Chl 506$_{PsbC}$ (Figs. S13f and S14f).

Chls *c* are mainly located at the periphery of PSII−ACPII (Fig. 4c). Our structural data and EET simulation indicate that Chls *c* transfer energy efficiently to Chls *a*, forming efficient EET pathways with surrounding Chls *a* and enhancing the blue-green light absorption of ACPIIs. Hence, our data support the proposed functions of Chl *c* in light harvesting[38]. Its potential role in energy dissipation merits further verification.

Cryptophytes and diatoms are abundant photosynthetic organisms in aquatic environments and thrive in different environmental niches. Diatoms possess a remarkable ability to adjust to changing levels of light and can flourish in both shallow and deep water, whereas the abundance of cryptophytes declines as the photic depth increases[43]. Diatoms contain a high content of Chls *c* and fucoxanthins associated with FCPs, which enable diatoms to absorb light in the blue-green region, enhancing their ability to survive in deep water. The amount of Chls *c* and Cars in cryptophyte ACPs are far less than that in diatom FCPs, which limits the ability of cryptophytes to absorb blue-green light dominant in deep water and may constrain their growth in deep water. However, this may be compensated somehow by the presence of phycobiliproteins that can absorb blue-green light. In addition, the diadinoxanthin-diatoxanthin (Ddx-Dtx) cycle in diatom FCPs quenches excess energy under strong light conditions in surface water, contributing to nonphotochemical quenching (NPQ)[44]. In contrast, the photoprotective Cars, such as alloxanthin, are enriched in cryptophytes, allowing cryptophytes to thrive under high light in shallow water[45]. Overall, the specific pigment arrangement of ACPIIs leads to the formation of novel EET pathways in cryptophyte PSII−ACPII. The differences in EET pathways of cryptophyte PSII−ACPII and diatom PSII−FCPII due to their distinct LHC structures and arrangements, as well as their distinct pigment compositions, might provide the foundation for cryptophytes and diatoms to thrive in their specific environmental niches.

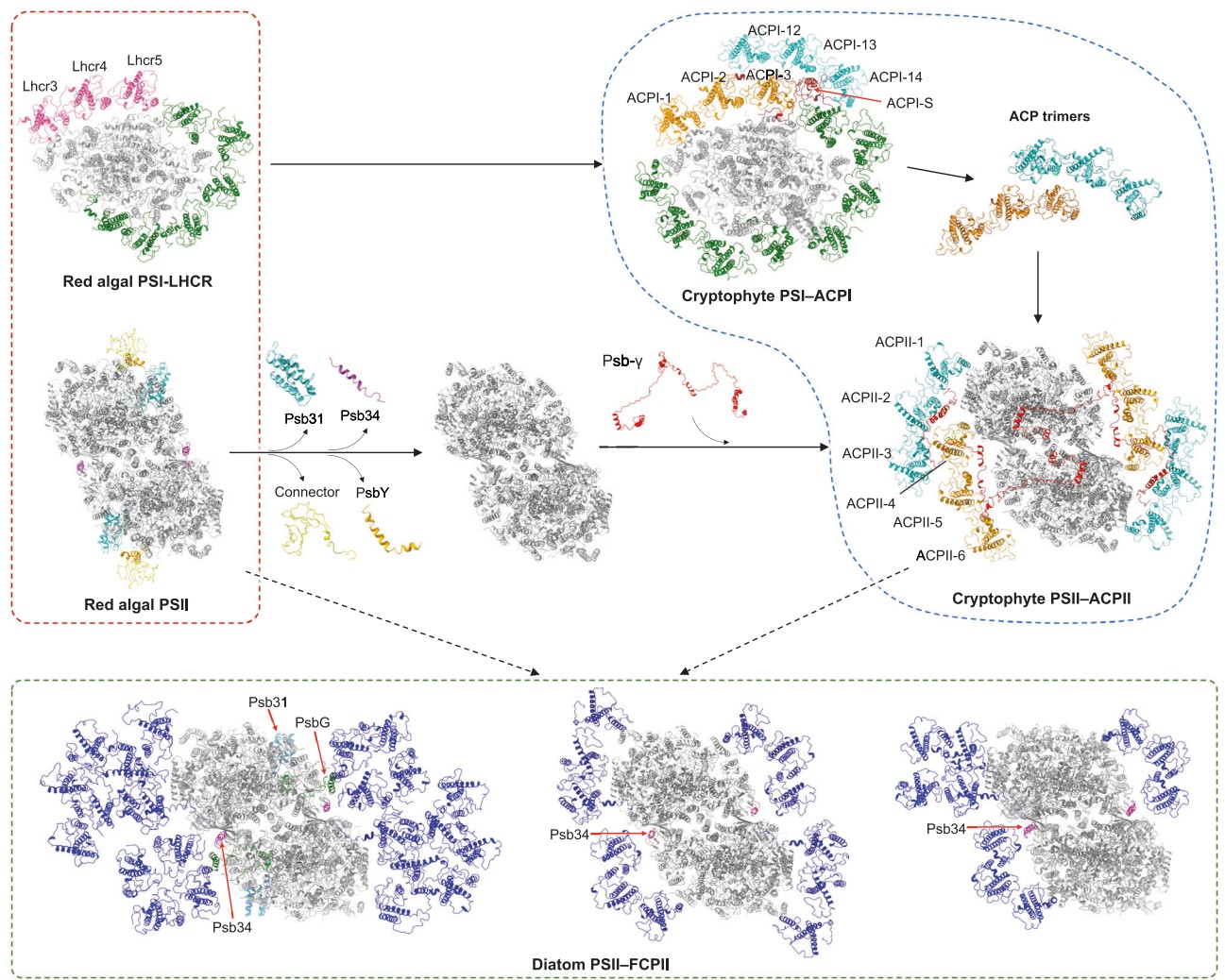

**Fig. 5 | Possible evolutionary development of red-lineage PSII–LHCII super-complexes.** Cryptophyte PSII–ACPII represents an intermediate structure between red alga PSII and diatom PSII–FCPII. During secondary endosymbiosis from red algae to cryptophytes, it is presumed that the connector and linker proteins were discarded, leading to the loss of phycobilisomes, Psb31, PsbY, and Psb34. The evolution of Psb-γ in cryptophytes facilitates the formation of the ACP trimeric assemblies and their association with the PSII dimeric core. ACPIIs might have evolved from ACPIs, and both are thought to originate from red algae LHCRs and can dynamically migrate between PSI and PSII in cryptophytes during state transitions. The PSII supercomplexes of red algae, cryptophytes, and diatoms exhibit various antenna organizations and overall architectures, highlighting the evolutionary diversification of PSII–LHCII in red lineages. Color code: red alga PSI (Lhcr3/ 4/5: pink; other Lhcr: green, core: gray); red alga PSII (Psb31: cyan; Psb34: magenta; Connector: yellow; PsbY: orange; other core subunits: gray); cryptophyte PSI–ACPI (ACPI-1/2/3: orange; ACPI-12/13/14: cyan; ACPI-S: red; other ACPI: green; PSII core: gray); cryptophyte PSII–ACPII (ACPII-1/2/3: cyan; ACPII-4/5/6: orange; ACPII-γ: red; PSII core: gray); diatom PSII–FCPII (FCPII: blue; Psb31: cyan; Psb34: magenta; PsbG: green; other PSII core subunits: gray). Structures used: cryptophyte PSII–ACPII (PDB: 8XR6), cryptophyte PSI–ACPI (PDB: 7Y7B), red alga PSI and PSII (PDB: 7Y5E), and diatom PSII–FCPII (Left, PDB: 6JLU, from *Chaetoceros gracilis*, containing PsbG, Psb31, and 22 FCPIIs; middle, PDB: 8J5K, from *Cyclotella meneghiniana*, containing 12 FCPIIs; right, PDB: 8IWH, from *Thalassiosira pseudonana*, containing 10 FCPIIs.

## Insights into the evolution of red lineage PSII–LHCII

Our structural data show that cryptophyte PSII–ACPII shares common architectural features with red algal PSII and diatom PSII–FCPII, while also displaying unique features. The structural similarities and variations determined in red-lineage PSII supercomplexes highlight the intermediate state of cryptophyte PSII in the evolution of red-lineage plastids (Fig. 5). This is in line with the findings of red lineage PSI–LHCI[30].

The remarkable differences among red-lineage oxyphototrophs are the type and architecture of LHCs. Red algae employ the extrinsic antenna phycobilisomes and membrane-integrated LHCs but lack Chl *c*, whereas diatoms possess Chl *a/c*-containing LHCs but do not have phycobilisomes and phycobiliproteins. In contrast, cryptophytes utilize both phycobiliproteins and alloxanthin Chl *a/c*-binding LHCs to extend the spectral region of light capturing[34,35,46–51].

Our structural analysis reveals that both PSII of cryptophytes and diatoms possess membrane-integrated Chl *c*-containing LHCIIs, which are absent in red algal PSII. Cryptophytes, being the earliest group that originated from red algae ancestors[26–28], were the first oxyphototrophs in the red lineage to have these membrane-spanning LHCIIs. This is consistent with the striking structural similarity between cryptophyte ACPIIs and red algal LHCRs (Fig. S9b). During secondary endosymbiosis from red algae to cryptophytes, it is presumed that the connector and linker proteins ($L_{RC}3$, $L_{RC}2$, $L_{PP}1$, $L_{PP}2$) for the phycobilisomes–PSII association were discarded. This led to the loss of phycobilisomes, the extrinsic Psb31 subunit, as well as the PsbY and Psb34 subunits that interact with the connector and $L_{RC}3$, respectively. Subsequently, Psb-γ was evolved in cryptophytes, playing a role in facilitating the formation of two groups of ACP trimeric assemblies that resemble red algae LHCRs and their association with the PSII core

(Fig. 5). Given the structural similarity of the trimeric assemblies of both ACPI and ACPII (Fig. S9a), as well as their arrangements in the PSI−ACPI and PSII−ACPII supercomplexes (Fig. S11d), it is speculated that ACPIIs might have evolved from ACPIs, and both are thought to originate from red algae LHCRs and can dynamically migrate between PSI and PSII in cryptophytes during state transitions[50,52]. Compared to the FCPIIs from diatoms, the most recent major algae added to the geological record[53,54], cryptophyte ACPIIs exhibit different structures and arrangements in the PSII supercomplex, suggesting a distant relationship between cryptophyte ACPII and diatom FCPII. Like cryptophyte PSII−ACPII, various PSII−LHCII "intermediate" structures from different organisms may have evolved during secondary endosymbiosis from red algae to diatoms[55].

## Methods

### Purification of the PSII−ACPII

*Chroomonas placoidea* T11 (a gift from Prof. Chen Min, College of Chemical and Biological Sciences and Engineering, Yantai University, Shandong, China) and diatom *Chaetoceros gracilis* were grown in 4 L of F/2 medium at 23 °C under continuous light illumination of 40 μmol photon m$^{-2}$ s$^{-1}$ with bubbling of air. The cells at logarithmic phase were collected by centrifugation (8000 × *g*, 5 min) and washed with MES-1 (25 mM MES-NaOH, pH 6.5, 1.0 M betaine, 10 mM MgCl$_2$). The cells were disrupted with glass beads beads for *C. placoidea* and by one freeze-thawing cycle for *C. gracilis* in MES-1[56,57]. The cell debris and unbroken cells were removed by centrifugation at 3000 × *g* for 2 min, and the thylakoid membranes were collected by centrifugation at 21,000 × *g* for 20 min for *C. placoidea* and 100,000 × *g*, 45 min for *C. gracilis*. The membrane pellet was washed with MES-2 (25 mM MES-NaOH, pH 6.5, 1.0 M betaine, 1.0 mM EDTA), and solubilized with n-dodecyl-α-D-maltopyranoside (α-DDM) (Anatrace, USA) (Chl:α-DDM = 1:30, 15 min) for *C. placoidea* at 0.4 mg mL$^{-1}$ Chl and 0.7% α-DDM for *C. gracilis* at 1.0 mg mL$^{-1}$ Chl in MES-3 (25 mM MES, pH 6.5, 1.0 M betaine, 10 mM NaCl, 5.0 mM CaCl$_2$) with gentle stirring. The solubilized membranes were centrifuged at 21,000 × *g* for 20 min or *C. placoidea* and 40,000 × *g*, 12 min for *C. gracilis*, and the supernatant was loaded onto a sucrose gradient of 10−30% in MES-3 containing 0.02% α-DDM. After centrifugation at 230,500 × *g* for 20 h (Beckman SW41Ti rotor), the PSII−ACPII and PSII−FCPII samples were collected from the 22% layer. The samples were concentrated to 3.0 mg ml$^{-1}$ in Chl, and sucrose was removed with MES-4 (25 mM MES, pH 6.5, 0.5 M betaine, 50 mM NaCl, 5.0 mM CaCl$_2$, 0.02% α-DDM) using a 100 kDa cut-off filter (Amicon Ultra; Millipore) for cryo-EM specimen preparation. All steps were performed at 4 °C under dim light.

### Biochemical characterization of PSII−ACPII

The subunit composition of PSII−ACPII was identified by denaturing gradient gel electrophoresis and mass spectrometry analysis[30,58]. Samples were denatured and separated by 8−16% SDS-PAGE. The protein bands were extracted from the gel. Then the proteins were reduced, alkylated, and digested with dithiothreitol, iodoacetamide, and trypsin, respectively. The resulting peptide fragments were examined by liquid chromatography-tandem mass spectrometry (LC-MS/MS) (Supplementary Data 1). The peptides were separated by a phase trap column (nanoViper C18, 100 μm × 2 cm, Thermo Fisher) connected to the C18-reversed phase analytical column (75 μm × 10 cm, 3 μm resin, Thermo Fisher) using the Easy-nLC 1000 System. The separated peptides were detected by a Q Exactive mass spectrometer (Thermo Fisher). The acquired spectra were searched against the selected database using the MASCOT engine (version 2.4) with the Proteome Discovery searching algorithm (version 1.4).

Pigment composition of the PSII−ACPII was analyzed by high-performance liquid chromatography (HPLC) system equipped with a Shimadzu photodiode array detector. Pigments extracted from 30 μL concentrated sample by 120 μL pre-cooled 100% acetone overnight at

4 °C were injected into C18 reversed-phase column (Waters, Ireland). The pigment extraction and the elution procedure were performed as described previously[40,59]. The eluates were detected by photodiode array detector at 445 nm. The wavelength detection range is 300 to 800 nm. Six major peaks were found in elution profiles. The pigments were identified based on their characteristic absorption peaks, in accordance with previous reports[35,47,60].

Room-temperature absorption spectra were recorded with a Shimadzu UV−Vis 1990 spectrophotometer. Oxygen-evolving activity was measured by a Clark-type oxygen electrode (Hansatech, UK) under saturating light at 25 °C in a buffer containing 50 mM Mes-NaOH (pH 6.5), 0.4 M sucrose, 15 mM NaCl, 5 mM MgCl$_2$, and 5 mM CaCl$_2$ at 7.5 μg Chl ml$^{-1}$[61-63]. Phenyl-p-benzoquinone (0.5 mM) and potassium ferricyanide (III) (0.5 mM) were used as the electron acceptor. The oxygen-evolving activity of the purified PSII−ACPII was determined to be 128 ± 13 (mean ± SD) μmol O$_2$ (mg Chl)$^{-1}$ h$^{-1}$ from three replicates.

### Sequence analysis of PSII−ACPII

High-throughput sequencing of *C. placoidea* transcriptome was performed by BioMarker (BMK) following the previously reported procedure[40]. Total RNA was extracted from cells. The NEBNext Ultra RNA Library Prep Kit from Illumina (NEB, MA, USA) was used to construct the cDNA Library following the manufacturer's instructions. The mRNA fragments were used as templates for the first strand of cDNA synthesis with random hexamer primers and RNase H, and the second-strand cDNA was synthesized using DNA polymerase I and RNase H. The synthesized double-stranded cDNA was modified by terminal repair, A-tailing, and adapter addition for hybridization. The AMPure XP beads (Beckman Coulter, Beverly, USA) were used for the selection of cDNA fragments (~150 bp). PCR was performed to obtain the final cDNA library after the cDNA was treated with USER Enzyme (NEB). The Illumina HiSeq 2000 platform were adopted to sequence the library preparations after clustering of the index-coded samples. Then the paired-end reads were generated. Based on the left.fq and right.fq, the transcriptome was assembled de novo using Trinity[64]. The transcriptome sequences were used for identifying the sequences of the PSII core and ACPII subunits by retrieving the homologous sequences against the transcriptome. The sequence alignment was done using CLC Sequence Viewer 8.0 and ESPript 3.0. The sequences for producing the phylogenetic tree were aligned with MUSCLE (default parameters), and the phylogenetic tree was constructed using MEGA X[65].

### Cryo-EM data collection and processing

The PSII−ACPII sample (4.0 μL, 2.0 mg mL$^{-1}$) was applied onto the freshly prepared grids (Quantifoil Au R2/1, 200 mesh) using a Vitrobot Mark IV (Thermo Fisher Scientific) operating at 100% humidity and 4 °C. Data collection was conducted using EPU (Thermo Fisher Scientific) on a 300 kV Titan Krios G3$^i$ microscope (Thermo Fisher Scientific) equipped with a K3 BioQuantum direct electron detector (Gatan Inc.) with a 20 eV energy filter slit (GIF, Gatan). All images were recorded at a nominal magnification of 81,000, resulting a pixel size of 0.53 Å and a total dose of 50 e$^-$ Å$^{-2}$. A total of 5382 movies were recorded and the defocus value was set from −1.2 to −2.2 μm.

Images were processed mainly by cryoSPARC v3.3.1[66]. For details, all movies were patch motion corrected and binned by a factor of 2 with dose weighting. Followed by Patch CTF Estimation and automatic particle picking. After particle extraction and two rounds of reference-free 2D classifications, 385,767 particles were selected for Ab-initial reconstruction and 3D classification with C1 symmetry. Two sets with 330,218 particles were selected for homogeneous refinement and achieving a map of 2.8 Å with C1 symmetry. To improve the resolution of the density map, further several rounds of 3D classification were performed and a final set of 168,683 were subjected to 3D non-uniform refinement and sharpening, global (per-group) CTF refinement and local (per-particle) CTF refinement. Then, 3D variability analysis was

performed by using cryoSPARC v3.3.1, followed by particle subtraction and local refinement with a soft mask for the regions of the peripheral ACPIIs. The overall resolutions of the map were 2.53 Å with C2 symmetry based on the gold-standard FSC 0.143.

## Model building and refinement

For model building of the cryptophyte PSII–ACPII supercomplex, the PSII core of diatom PSII–FCPII (PDB ID: 6JLU)[20], the cryptophyte ACPI-1/2/3 trimer and the cryptophyte ACPI-12/13/14 trimer were first rigid body fitted into the 2.53 Å cryo-EM map using UCSF Chimera[67]. ACPI-1/2/3 trimer and ACPI-12/13/14 trimer were fitted into the map regions of ACPII trimer close to the core and distant from the core, respectively. The sequences of the *C. placoidea* PSII core identified in the transcriptome were used for correcting the amino acid residues of each chain. The sequences of PsbW and subunit Psb-γ were searched from the transcriptome using the estimated sequences based on the cryo-EM map as previously described[40]. The potential sequences of ACPIIs were searched from the transcriptome using the sequences of ACPI-1/2/3/12/13/14. ACPIIs were identified based on the best match of the amino acid sequence with the cryo-EM map. The Psb-γ subunit was constructed by de novo model building with COOT[68]. Chl *a* and Chl *c* were distinguished by the density map corresponding to the phytol chain for Chl *a*, and the planarity of C-18[1], C-18, C-17, and C-17[1] resulting from the C-18 = C-17 double bound for Chl *c*[30,69,70]. Alloxanthin was assigned based on the density covering the two end groups shown with a threshold of 12 σ contour level as alloxanthin has more areas for the hydroxy (Fig. S3c). As alloxanthin and monadoxanthin cannot be distinguished based on the current resolution of the density map, the possible binding sites for monadoxanthin were assigned as alloxanthin. ACPIIs share similar structural features with ACPIs that associate with cryptophyte PSI, and the pigment-binding sites are largely conserved among ACPIIs and ACPIs[30]. Furthermore, ACPII-1/2/3 and ACPII-4/5/6 form LHCII trimers, resembling the ACPI-14/13/12 and ACPI-3/2/1 trimers in cryptophyte PSI–ACPI (Figs. S9a, S11d). Therefore, the pigment assignment to the area with lower local resolution in ACPIIs was referred to those in ACPIs, and the determination of pigments in ACPII-4, ACPII-5, and ACPII-6 which have lower map densities referred to the pigments in ACPI-3, ACPI-2, and ACPI-1, respectively. The Chl *c*310 in ACPII-4 was assigned on the basis of the identical Chl *c* sites in ACPII-1 since ACPII-4 and ACPII-1 share similar structures and their Chls *c*310 share comparable orientations. The Grade Web Server was adopted to generate geometrical restraints of pigments. All the residues and cofactors were manually adjusted with COOT. The incorporation of water molecules around the Oxygen-Evolving Complex (OEC) area was performed manually using Coot, based on the density map and previously reported PSII structure (PDB ID: 8IR5)[71]. Phenix was adopted to refine the constructed model via real-space refinement[72]. Manual correction and adjustment were performed with COOT to resolve atomic clashes and geometry problems after real-space refinement. Manual correction and Phenix refinement were carried out iteratively to reach the final atomic model. The geometries of the structural model were evaluated with Phenix and the statistics was summarized in Table S1.

## Simulation of the excitation energy transfer in PSII–ACPII

Excitation energy transfer within PSII–ACPII was studied in the limit of the Förster theory as previously described[40,41]. The energy transfer rates between Chls, *T* values in ps$^{-1}$, were calculated using the following equation:

$$T_{mn} = \frac{2\pi}{\hbar} \left| V_{mn} \right|^2 \int_{-\infty}^{\infty} d\omega F_m(\omega) A_n(\omega) \tag{1}$$

$V_{mn}$ represents the electron coupling strength between pigments. It was calculated using the TrEsp (transition charges from electrostatic potentials) method[73] and Multiwfn program[74], which incorporate environmental screening factors[75], were used to calculate $V_{mn}$.

$$V_{mn} = f \cdot \sum_{l \in m, l' \in n}^{L} \frac{q_l^T \cdot q_{l'}^T}{\left| R_{l,l'} \right|} \tag{2}$$

$$f = \begin{cases} 1; R \le 6.6 \\ A \cdot \exp(-\beta R) + 0.54; 6.6 < R < 20 \\ 0.54; 20 \le R \end{cases} \tag{3}$$

$l$ and $R_{l,l'}$ represents the $l$-th atom and the distance between the corresponding atoms of the pigment molecule $m$, respectively. $q^T$ denotes the transition charge of the atom, and $f$ is the environmental screening factor.

$F_m(\omega)$ and $A_n(\omega)$ represent the emission spectrum of pigment $m$ and the absorption spectrum of pigment $n$, respectively. These spectral properties of pigments were simulated by a Gaussian broadening function[76]:

$$F_m(\omega) = \frac{1}{\sigma_m \sqrt{2\pi}} \exp\left\{ -\frac{(\omega_m - S - \omega)^2}{2\sigma_m^2} \right\} \tag{4}$$

$$A_n(\omega) = \frac{1}{\sigma_n \sqrt{2\pi}} \exp\left\{ -\frac{(\omega_n - \omega)^2}{2\sigma_n^2} \right\} \tag{5}$$

$S$ and $\sigma$ represents the Stokes shift and the full width at half maximum (FWHM), respectively.

In addition, the energy transfer rates between ACPIIs as well as between ACPII and PSII core pigment aggregates were simulated by the generalized Förster theory, an extension of the classical Förster theory[42]:

$$k_{DA} = \frac{2\pi}{\hbar} \sum_{\alpha \in D} \sum_{\beta \in A} \frac{\exp\left(-\frac{\varepsilon_\alpha^D}{k_B T}\right)}{Z} \left| V_{\alpha\beta}^{DA} \right|^2 \int d\omega S_\alpha^D(\omega) S_\beta^A(\omega) \tag{6}$$

D and A refers to the donor and acceptor aggregate, respectively. $\alpha$ represents the exciton state index of the donor aggregate, while the $\beta$ denote the exciton state index of the acceptor aggregate. $\varepsilon$ represents the eigenvalue of the Hamiltonian for the donor or acceptor aggregate. $Z$ is the partition function, and represents the electronic coupling between excitonic states of the donor and acceptor aggregates.

$$Z = \sum_\alpha \exp\left(-\frac{\varepsilon_\alpha^D}{k_B T}\right) \cdot V_{\alpha\beta}^{DA} \tag{7}$$

The above parameters were calculated by Gaussian16 software (Gaussian, Inc. Wallingford, CT, USA) and custom Python scripts (https://doi.org/10.5281/zenodo.10791187). CAM-B3LYP/6-31G*[77] was used and the keyword Iop (9/40 = 4) was added to the calculation to print out more detailed configuration coefficients for the subsequent calculation of electronic coupling. The CAM-B3LYP function was selected as it has been reported to offer a good description of excited states for chlorophyll-like systems[78–81]. When simulating the spectrum, the S and FWHM of Chl *a* were 160 cm$^{-1}$ and 240 cm$^{-1}$[82]. In addition, we selected the porphyrin rings of Chl molecules (Fig. S15) for excitation spectra calculation with time-dependent density functional theory (TDDFT)[83], and the spectral overlap between Chl *c* and Chl *a* was replaced with experimental data. The electronic coupling term was calculated by using the TrEsp method[73] following the Eq. (2).

## Reporting summary

Further information on research design is available in the Nature Portfolio Reporting Summary linked to this article.

## Data availability

The cryo-EM map and atomic coordinates have been deposited in the Protein Data Bank and the Electron Microscopy Data Bank under the accession numbers 8XR6 and EMD-38596, respectively. The atomic coordinates data used in this study are available in the Protein Data Bank database under the accession codes 4YUU, 6KAF, 6JLU, 7VD5, 7Y5E, 7Y7B, 8IR5, 8J5K, 8IWH. The RNA-seq data have been deposited in the NCBI Sequence Read Archive (SRA) database under the accession code PRJNA1120208. Source data for Supplementary Fig. 1b, 1c, and 1d are provided in the Source data file. Source data are provided with this paper.

## Code availability

The custom Python scripts used in this study are available in GitHub [https://doi.org/10.5281/zenodo.10791187].

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

## Acknowledgements

We thank Yi Zhang (College of Marine Life Sciences, Ocean University of China) and Fang Zhao (State Key Laboratory of Microbial Technology, Shandong University, Qingdao, China) for data analysis. We thank Mei-Ling Sun (College of Marine Life Sciences, Ocean University of China) and Xiao-Ju Li (State Key Laboratory of Microbial Technology, Shandong University, China) for their help with TEM. We are grateful to Ceng Gao and Xiao-Wei Li (Cryo-EM facility for Marine Biology, Laoshan laboratory) for cryo-EM data collection. Numerical computations were performed on Hefei Advanced Computing Center. This work was supported by National Key R&D Program of China (2023YFA0914600 to L.-S.Z. and L.-N.L., 2021YFA0909600 to L.-N.L.), Marine S&T Fund of Shandong Province for Qingdao Marine Science and Technology Center (2022QNLM030004-3), National Natural Science Foundation of China (91851205 to Y.-Z.Z., 32100028 to K.L., 32100200 to LSZ, 32070109 to L.-N.L., 21873034 to J.G.), Royal Society (URF\R\180030, IEC\NSFC\191600 to L.-N.L.), Biotechnology and Biological Sciences Research Council (BBSRC) (BB/V009729/1, BB/R003890/1, and BB/W001012/1 to L.-N.L.), and Fundamental Research Funds for the Central Universities (2662024XXPY003 and 2662023XXPY006).

## Author contributions

Y.-Z.Z., L.-S.Z., and L.-N.L. conceived the project; L.-S.Z., B.-Y.Q., and Q.-B.Z. performed the sample preparation and characterization; K.L., and D.-L.Z. collected the cryo-EM data. K.L. processed the cryo-EM data and reconstructed the cryo-EM density map. K.L., L.-S.Z., B.-Y.Q., and Q.-B.Z. built the structure model and refined the structure. J.-P.G. and J.G. performed computational simulations for EET. Y.-Z.Z., K.L., L.-S.Z., X.-L.C., and L.-N.L. analyzed the data. Y.-Z.Z., K.L., L.-S.Z., X.-L.C., and L.-N.L. wrote the manuscript paper with contributions from all other authors.

## Competing interests

The authors declare no competing interests.
