## [Peer Review File · Nature Communications]

Reviewers' Comments:

Reviewer #1:

Remarks to the Author:

This article presents a cryo-EM structure of the PSII-ACPII supercomplex from the cryptophyte *Chroomonas placoidea*.

I find the overall style to be good, both presentation, reported methods, reproducibility and language. The article can be published pending some minor changes. Some remarks are given below.

The introduction draws an analogy between all PSII systems, by claiming them to form "PSII-LHCII supercomplexes". Phycobilisomes are usually not classified as LHCII proteins, which prevents such a generic classification when also including cyanobacteria.

The analysis of the structure, especially of the Psb- γ protein, is interesting; the latter likely represents the most important finding of the article. A direct mapping of the PDB chain names to the chain names in the paper would have however been helpful, if only mentioning "chains N and n" in the context of Psb- γ , for example.

The EET analysis yields the expected results of strong coupling between the complexes. However, the corresponding method section lacks the references for the employed quantum chemistry (CAM-B3LYP and the 6-31G* basis). It is also required that some articles are cited which have used the combination of CAM-B3LYP/6-31G* for light harvesting systems before - otherwise, the choice of methods is arbitrary and would require benchmarking, which is not the article's scope. Please give the appropriate credit to previous computational work just as you would do for experimental work.

List of minor remarks:

Abstract: "primitive" is an inappropriate way to classify an organism. Please use a synonym that does not suggest inferiority compared to "higher/advanced" (the opposite of "primitive") organisms.

Fig. 5: "Con(n)ector"

Reviewer #2:

Remarks to the Author:

The manuscript by Zhang et al. describes a high-resolution cryoEM structure of photosystem II from the cryptophyte *Chroomonas placoidea*. They found that cryptophyte PSII shows an unusual arrangement of ACPII complexes as linear trimers. They also uncovered a novel pigment binding subunit termed Psb- γ , connecting ACPII to PSII core. The manuscript discusses the evolutionary aspect of PSII across algae. Although high-resolution structures of red alga and diatom PSII were reported previously, the current work shows novel features of PSII and undisclosed binding mode of the antennae to the core complex. The data presented here is solid, however in my view the manuscript and the data analysis need to be improved before considering it for publication.

Major comments

The oxygen evolution activity needs to be presented in the manuscript or in the supplementary data to demonstrate the active complex was purified.

How were the locations of chlorophyll c2 and alloxanthins determined? Many of the map densities are not conclusive. Have you used any supporting analysis to determine their positions in the ACPIIs.

Structural flexibility – have you tried 3Dflex analysis, focused refinement, or multibody refinement? Is it possible there are additional compositional states that were not separated during classification? These would be important for ACPII-4/5/6 which have lower quality map densities.

Have you applied symmetry in the refinement process? If so, please state where it was applied in the data processing workflow and if C1 yielded a similar map.

There are many densities close to the OEC but there are no water molecules present in the model. At this resolution water molecules should be identified. Please add these to the model and compare the water network with other PSII structures from various origins.

QB was modelled in although the density does not show it is there definitively, but nonetheless it is very interesting to see it there. Can you compare it with other structures where QB is present? (Diatoms, green algae, cyanobacteria)

PsbW is a single transmembrane subunit connecting the core with LHC subunits. Its location in diatom PSII is identical to PsbW in green algae (PDB 6KAF for example). Please discuss their evolutionary relationship.

Lines 152-153 – there is also a lower capacity in the orange absorption spectra (500-550nm) and a slight red-shift in the red absorption. Is this the result of chlorophyll c2 or alloxanthin and how it coincides with chlorophyll c2 blue-shift in the red absorption spectra? Please discuss the potential effect of these pigments on excitation energy transfer in PSII-ACPII.

Lines 153-157 - the Chl-binding sites are largely conserved among all ACPIIs – please add a comparison with green algae LHCII sites for both chlorophylls and carotenoids.

Possible energy transfer pathways within the PSII-ACPII supercomplex – the analysis needs to be improved by adding exact distances when comparing the pigment locations with *Chaetoceros gracilis* and *Cyclotella meneghiniana*. I did not find any reference to the EET contribution of chlorophyll c2, its energy transfer coefficient and a proposed role for its presence in the ACPII complexes.

Modelling partially incorrect (CLA/B/617; Lhc4/5/6/0/p/g) and map densities are incomplete, especially in ACPII-6. Please improve the map and revise the model.

Minor comments

Line 25 – ..in light harvesting...

Line 107 – “Psb-γ has a single transmembrane (TM) α-helix (A19-A45)”: in the model the residue numbers is A82-A108, please clarify which is correct.

Line 129 – “However, there are variations in the location and orientation of the β-turns (K63-E71)”: in the model the residue numbers is K101-E109, please clarify which is correct.

Line 155 – “The Chl 314/315 were close to the PSII core” – please note what is the shortest distance of these to the core pigments.

Line 160 – “..BC loop and the N-terminal loop..” – please state these are ACPII loops.

Line 235 – “..might provide the foundation for cryptophytes and diatoms to thrive in their specific environmental niches”. Please add a short explanation on how different their niches are and what would be the added value of their unique pigments.

Line 402 – which chlorophyll atoms were used to calculate the vector of the spectral overlap?

Figure 5 – please add the subunits marked in the figure and their colors to the legend. In the figure should be “Connector” and not “Conector”.

Supplementary table 1 – please add the map vs model CC.

Responses to Reviewers' Comments

Reviewer #1:

This article presents a cryo-EM structure of the PSII-ACPII supercomplex from the cryptophyte *Chroomonas placoidea*. I find the overall style to be good, both presentation, reported methods, reproducibility and language. The article can be published pending some minor changes. Some remarks are given below.

Reply: We sincerely appreciate Reviewer 1's extremely positive comments on our work.

1. The introduction draws an analogy between all PSII systems, by claiming them to form "PSII-LHCII supercomplexes". Phycobilisomes are usually not classified as LHCII proteins, which prevents such a generic classification when also including cyanobacteria.

Reply: Thanks for pointing this out. We have revised the relevant descriptions in INTRODUCTION to distinguish between phycobilisomes and LHCII (Page 3).

2. The analysis of the structure, especially of the Psb- γ protein, is interesting; the latter likely represents the most important finding of the article. A direct mapping of the PDB chain names to the chain names in the paper would have however been helpful, if only mentioning "chains N and n" in the context of Psb- γ , for example.

Reply: Thanks for the suggestion. We have added a direct mapping of the PDB chain names to the chain names in the paper in Table S3. The chain names in the paper have been labelled by the PDB chain names in the revised manuscript.

3. The EET analysis yields the expected results of strong coupling between the complexes. However, the corresponding method section lacks the references for the employed quantum chemistry (CAM-B3LYP and the 6-31G* basis). It is also required that some articles are cited which have used the combination of CAM-B3LYP/6-31G* for light harvesting systems before - otherwise, the choice of methods is arbitrary and would require benchmarking, which is not the article's scope. Please give the appropriate credit to previous computational work just as you would do for experimental work.

Reply: We have added the reference (Yanai et al., *Chem. Phys. Lett.* 2004, 393: 51–57) for the employed quantum chemistry and cited previous articles which have used this quantum chemistry method to make the description and discussion more rigorous (Page 17).

List of minor remarks:

4. Abstract: "primitive" is an inappropriate way to classify an organism. Please use a synonym that does not suggest inferiority compared to "higher/advanced" (the opposite of "primitive") organisms.

Reply: We have replaced "primitive" with "ancestral".

5. Fig. 5: "Con(n)ector"

Reply: We have corrected the typo in Fig. 5.

Reviewer #2 (Remarks to the Author):

The manuscript by Zhang et al. describes a high-resolution cryoEM structure of photosystem II from the cryptophyte *Chroomonas placoidea*. They found that cryptophyte PSII shows an unusual arrangement of ACPII complexes as linear trimers. They also uncovered a novel pigment binding subunit termed Psb- γ , connecting ACPII to PSII core. The manuscript discusses the evolutionary aspect of PSII across algae. Although high-resolution structures of red alga and diatom PSII were reported previously, the current work shows novel features of PSII and undisclosed binding mode of

the antennae to the core complex. The data presented here is solid, however in my view the manuscript and the data analysis need to be improved before considering it for publication.

Reply: We sincerely appreciate Reviewer 2's highly positive comments on our work.

Major comments

1. The oxygen evolution activity needs to be presented in the manuscript or in the supplementary data to demonstrate the active complex was purified.

Reply: We have measured the oxygen-evolving activity to verify the active state of the purified PSII–ACPII supercomplexes. The oxygen-evolving activity of purified PSII–ACPII was $128 \pm 13 \mu\text{mol O}_2 (\text{mg Chl})^{-1} \text{h}^{-1}$ from three replicates, comparable to the activities of reported PSII complexes (Wang et al., *Biochemistry*, 2020, 59(30):2823-2831; Kunath et al., *Photosynth Res*, 2012, 111:173-183; Wei et al., *Nature*, 2016, 534(7605):69-74; Sheng et al., *Nat Plants*, 2019, 5(12):1320-1330; Zhao et al., *Nat Commun*, 2023, 14(1):8164). We have added relevant descriptions in the main text (Page 4) and MATERIALS AND METHODS of the revised manuscript (Page 13).

2. How were the locations of chlorophyll c2 and alloxanthins determined? Many of the map densities are not conclusive. Have you used any supporting analysis to determine their positions in the ACPIIs.

Reply: As described in METHODS (Page 15), Chl *c* was assigned based on the absence of density map corresponding to the phytol chain for Chl *a* as well as the planarity of C-18¹, C-18, C-17, and C-17¹ resulting from the C-18=C-17 double bond for Chl *c*. Alloxanthin was assigned based on the density covering the two end groups shown with a threshold of 12 σ contour level as alloxanthin has more areas for the hydroxy (Fig. S3c).

ACPIIs share similar structural features with ACPIs associated with cryptophyte PSI, which we have recently determined at 2.7-Å resolution (*Plant Cell*, 2023, 35(7):2449-2463). The pigment binding sites are largely conserved among ACPIIs and ACPIs. Furthermore, ACPII-1/2/3 and ACPII-4/5/6 form LHCI trimers, resembling the ACPI-14/13/12 and ACPI-3/2/1 trimers in cryptophyte PSI–ACPI (Figs. S9a, S11d). Therefore, the pigment assignment to the area with lower local resolution in ACPIIs was referred to those in ACPIs, and the determination of pigments in ACPII-4, ACPII-5, and ACPII-6 which have lower map densities referred to the pigments in ACPI-3, ACPI-2, and ACPI-1, respectively. Chl *c*310 in ACPII-4 was assigned on the basis of the identical Chl *c* sites in ACPII-1, since ACPII-4 and ACPII-1 share similar structures and their Chls *c*310 share comparable orientation. We have added relevant descriptions in MATERIALS AND METHODS (Page 15) of the revised manuscript.

3. Structural flexibility – have you tried 3Dflex analysis, focused refinement, or multibody refinement? Is it possible there are additional compositional states that were not separated during classification? These would be important for ACPII-4/5/6 which have lower quality map densities.

Reply: We thank the reviewer for pointing out this important question. To address this question, we performed 3D variability analysis. The particles from different clusters were combined, and then particle subtraction was carried out, followed by local refinement with a focus on the peripheral subunits. We conducted analysis with masks of different sizes and positions for local search, particularly focusing on ACPII-6. By implementing local refinement, the resolution of the peripheral regions of the map was enhanced, resulting in an overall improvement in the quality of the structural model. The results revealed that the peripheral ACPII subunits exhibit a high degree of orientational flexibility, highlighting the dynamic architecture of PSII–ACPII. The data process and findings are illustrated in Figure R1 below, and have also been added in the new Fig. S2, in the main text (Page 8), and MATERIALS AND METHODS (Page 14) in the revised manuscript.

Fig. R1. 3D variability analysis and Local refinement of the peripheral subunits of the PSII-ACPII map. The gold standard Fourier shell correlation (FSC) curves of the density maps of with the criterion of 0.143. **a**, Representative cryo-EM images of the 3D variability analysis of peripheral subunits of PSII-ACPII. The images below show the perspective obtained by rotating the images above by 90°. **b**, The workflow diagram for local refinement.

4. Have you applied symmetry in the refinement process? If so, please state where it was applied in the data processing workflow and if C1 yielded a similar map.

Reply: We conducted Ab-Initio Reconstruction and Heterogeneous 3D Refinement with C1 symmetry, followed by Homogeneous Refinement with C2 symmetry. We have revised the data processing workflow as shown in Fig. S2. Additionally, we performed Homogeneous Refinement with C1 symmetry and compared the results in Chimera (Fig. R2 below). In the C1 symmetry map, the density appears slightly better at one end. However, in the C2 symmetry map, the averaging process resulted in consistent density across both parts of the map. Nevertheless, the two maps exhibit considerable overlap, indicating their similarity.

Fig. R2. Comparison of the map reconstructed with C1 and C2 symmetry. Blue, map with C1 symmetry. Salmon, map with C2 symmetry.

5. There are many densities close to the OEC but there are no water molecules present in the model. At this resolution water molecules should be identified. Please add these to the model and compare the water network with other PSII structures from various origins.

Reply: We tried automated addition of water molecules using software, but the map density was not sufficient to support this process. Hence, we manually incorporated water molecules into the Oxygen-Evolving Complex (OEC) area using Coot, based on the density and the comparison with reported PSII structures from cyanobacteria, red algae, and diatoms (Fig. R3, Fig. S4d). The water molecules have been added to the final structural model. We found that most cryo-EM structures could not provide sufficient density resolution for the identification of water molecules, whereas crystal structures provided better insights into water molecule distribution. Overlay comparisons reveal that the positions of OEC are conserved, and the distributions of most water molecules around OEC in the models are also similar, supporting the high conservation of the PSII core. We have added relevant descriptions in RESULTS AND DISCUSSION (Page 5) of the revised manuscript.

Fig. R3. Comparison of the locations and structures of Water molecules in cryptophyte PSII (red) with those in PSII of the cyanobacterium *Thermostrictus vulcanus* (green, PDB: 8IR5), red alga *Cyanidium caldarium* (blue, PDB: 4YUU), diatom *Chaetoceros gracilis* (yellow, PDB: 7VD5). Representative areas have been encircled to illustrate the relative positions between the two images.

6. QB was modelled in although the density does not show it is there definitively, but nonetheless it is very interesting to see it there. Can you compare it with other structures where QB is present? (Diatoms, green algae, cyanobacteria)

Reply: We have compared the locations and structures of Q_A and Q_B in cryptophyte PSII with those in PSII–LHCII of cyanobacteria, red alga, green alga, and diatom (Fig. R4, Fig. S4c). The locations and structures of Q_A and the head group of Q_B are highly conserved. In contrast, the tails of Q_B possess diverse conformations and orientations, suggesting the conformational flexibility of Q_B compared to Q_A in different PSII–LHCII structures. We have added relevant descriptions in RESULTS AND DISCUSSION (Page 4-5) of the revised manuscript.

Fig. R4. Comparison of the locations and structures of Q_A and Q_B in cryptophyte PSII (red) with those in PSII of cyanobacteria *Thermotichus vulcanus* (blue, PDB: 8IR5), red alga *Porphyridium purpureum* (cyan, PDB: 7Y5E), green alga *Chlamydomonas reinhardtii* (green, PDB: 6KAF), diatom *Chaetoceros gracilis* (magenta, PDB: 7VD5), and diatom *Cyclotella meneghiniana* (grey, PDB: 8J5K).

7. PsbW is a single transmembrane subunit connecting the PSII core with LHC subunits. Its location in diatom PSII is identical to PsbW in green algae (PDB 6KAF for example). Please discuss their evolutionary relationship.

Reply: We have compared the sequences of PsbW subunits in the PSII structures of cryptophyte, red algae, diatom, and green algae (Fig R5a, Fig. S6d). The result showed that PsbW of green algal PSII has a low sequence identity with those of red algal PSII–LHCII, cryptophyte PSII–ACPII, and diatom PSII–FCPII. Phylogenetic analysis revealed that green algal PsbW exhibits a distant evolutionary relationship with PsbW of red algae, cryptophytes, and diatoms (Fig. R5b, Fig. S6e). We have added relevant descriptions in RESULTS AND DISCUSSION (Page 6) of the revised manuscript.

Fig. R5. Sequence alignments and phylogenetic analysis of PsbW from cryptophytes, red algae, diatoms and green algae. a, Comparison of the sequence of PsbW from *Chroomonas placoidea* (C.p.) in this study and the sequences of PsbW in the PSII structures of red algae *Porphyridium purpureum* (P.p.), diatom *Chaetoceros gracilis* (C.g.), and green algae *Chlamydomonas reinhardtii* (C.r.). b, Phylogenetic tree of PsbW from PSII structures of cryptophyte *Chroomonas placoidea* (C.p.), red algae *Porphyridium purpureum* (P.p.), diatom *Chaetoceros gracilis* (C.g.), green algae *Chlamydomonas reinhardtii* (C.r.) and their homologous sequences. *Cryptophyta* sp. CCMP2293: C.sp., *Guillardia theta* CCMP2712: G.t., *Rhodorus marinus*: R.m., *Porphyridium purpureum*: P.p., *Galdieria yellowstonensis*: G.y., *Galdieria sulphuraria*: G.s. *Galdieria partita*: G.pa., *Chaetoceros gracilis*: C.g., *Thalassiosira pseudonana*: T.p., *Chlamydomonas reinhardtii*: C.r., *Chlamydomonas schloesseri*: C.s., *Gonium pectoral*: G.pe., *Pleodorina starrii*: P.s.. The phylogenetic tree was constructed using Neighbor-Joining method based on amino acid sequences. The tree was built with the Poisson model using 133 amino acid residues, and a bootstrap test (1000 replicates) was conducted.

8. Lines 152-153 – there is also a lower capacity in the orange absorption spectra (500-550nm) and a slight red-shift in the red absorption. Is this the result of chlorophyll c2 or alloxanthin and how it coincides with chlorophyll c2 blue-shift in the red absorption spectra? Please discuss the potential effect of these pigments on excitation energy transfer in PSII-ACP II.

Reply: The 12 ACP IIs in cryptophyte PSII-ACP II contain a total of 133 Chl *a*, 14 Chl *c*, and 60 Car (48 alloxanthin, 8 crocoxanthin, and 4 α -carotene) molecules. In contrast, PSII-FCP II of diatom *Chaetoceros gracilis* contains 22 FCP II antennae, binding in total 142 Chl *a*, 70 Chl *c*, and 120 Car molecules. Interestingly, the quantities of Chl *a* in cryptophyte ACP IIs and diatom FCP IIs are similar, whereas the amount of Chl *c* in FCP IIs is five times that in ACP IIs, and the amount of Car in FCP IIs is two times that in ACP IIs. The reduced content of Chl *c* and Car molecules in ACP IIs results in the weak absorption of cryptophyte PSII-ACP II at 500-550 nm. The large amount of Chl *c* in FCP IIs may lead to the blue shift of diatom PSII-FCP II in the red absorption spectra compared to PSII-ACP II.

The Chl *c* and Car pigments enable cryptophyte PSII-ACP II absorbs light in the blue-green region which could not be effectively absorbed by Chl *a*. In addition, cryptophytes possess phycobiliproteins, further enhancing their absorption of green light. These spectral features allow the survival of cryptophytes in deep water, where the blue-green light can penetrate. Moreover, Chls *c* could transfer energy efficiently to the coupled Chls *a* (Croce *et al.*, *Nat Chem Biol*, 2014, 10(7):492-501), and were proposed to facilitate energy transfer from Cars to Chls *a* (Larkum *et al.*, *Chlorophylls and Bacteriochlorophylls: Biochemistry, Biophysics, Functions and Applications*, 2006, pp. 261-282). Chl *c* has also been hypothesized to play a role in energy dissipation under high-light conditions (Tsuji-mur *et al.*, *J Phys Chem B*, 2023, 127(2):505-513). Thus, Chls *c* and Cars may form an energy-quenching system to protect PSII-ACP II from excess irradiation. We have added relevant descriptions in RESULTS AND DISCUSSION (Page 6-7) of the revised manuscript.

9. Lines 153-157 - the Chl-binding sites are largely conserved among all ACP IIs – please add a comparison with green algae LHC II sites for both chlorophylls and carotenoids.

Reply: We have added a comparison with green algal LHC II sites for both chlorophylls and carotenoids (Fig. R6, also Fig. S9 in the revised manuscript). Compared to green algal LHC IIs, cryptophyte ACP IIs have a very similar number of pigment binding sites, but the positions of some of these sites vary. Specifically, green algal LHC IIs have 14 conserved Chl-binding sites, whereas cryptophyte ACP IIs share the same positions with 9 of these. Moreover, green algal LHC IIs feature 4 conserved carotenoid-binding sites, whereas cryptophyte ACP IIs share 2 of them (sites 401, 403). We have added relevant descriptions in RESULTS AND DISCUSSION (Page 7) of the revised manuscript.

Fig. R6. Comparison of the LHCII of cryptophyte and green algae. Superposition of the binding sites for chlorophylls (a) and carotenoids (b) in cryptophyte ACPIIs (red) and LHCII of the green alga *Chlamydomonas reinhardtii* (green, PDB: 6KAF). The LHCII include CP26, CP29, and LHCII subunits from the S-, M- and N-LHCII trimers.

10. Possible energy transfer pathways within the PSII–ACPII supercomplex – the analysis needs to be improved by adding exact distances when comparing the pigment locations with *Chaetoceros gracilis* and *Cyclotella meneghiniana*. I did not find any reference to the EET contribution of chlorophyll c2, its energy transfer coefficient and a proposed role for its presence in the ACPII complexes.

Reply: We have added the distances into Fig. S13 and S14 as suggested by the reviewer. Chl *c* was proposed to play roles in both light harvesting and energy dissipation (Tsuji-mur *et al.*, *J Phys Chem B*, 2023, 127(2):505-513). In our study, the structural characterization revealed that Chls *c* are mainly located at the periphery of PSII–ACPII and form efficient EET pathways with surrounding Chls *a*, as shown in Fig. 4c. This suggests that Chls *c* could efficiently transfer energy to Chls *a*, enhancing the blue-green light absorption of ACPIIs. Our structural analysis and computational simulation support the proposed functions of Chl *c* in EET. We have added relevant descriptions in RESULTS AND DISCUSSION (Page 10) of the revised manuscript.

11. Modelling partially incorrect (CLA/B/617; Lhc4/5/6/0/p/g) and map densities are incomplete, especially in ACPII-6. Please improve the map and revise the model.

Reply: Thanks. We have improved the map regarding these regions and revised the model.

Minor comments

12. Line 25 – ..in light harvesting...

Reply: We have revised the sentence.

13. Line 107 – “Psb- γ has a single transmembrane (TM) α -helix (A19-A45)”: in the model the residue numbers is A82-A108, please clarify which is correct.

Reply: We have corrected the typo to A82-A108.

14. Line 129 – “However, there are variations in the location and orientation of the β -turns (K63-E71)”: in the model the residue numbers is K101-E109, please clarify which is correct.

Reply: We have corrected the typo to K129-E137.

15. Line 155 – “The Chl 314/315 were close to the PSII core” – please note what is the shortest distance of these to the core pigments.

Reply: The shortest distance of Chls 314/315 to Chl 617_{PsbB} within the PSI core are 4.1 nm and 6.3 nm, respectively. We have added relevant descriptions in RESULTS AND DISCUSSION (Page 7) of the revised manuscript.

16. Line 160 – “..BC loop and the N-terminal loop..” – please state these are ACPII loops.

Reply: Thanks. We have stated these in RESULTS AND DISCUSSION (Page 7) of the revised manuscript.

17. Line 235 – “..might provide the foundation for cryptophytes and diatoms to thrive in their specific environmental niches”. Please add a short explanation on how different their niches are and what would be the added value of their unique pigments.

Reply: We have added explanations in RESULTS AND DISCUSSION (Page 10) of the revised manuscript. “Cryptophytes and diatoms are abundant photosynthetic organisms in aquatic environments and thrive in different environmental niches. Diatoms possess a remarkable ability to adjust to changing levels of light and can flourish in both shallow and deep water, whereas the abundance of cryptophytes declines as the photic depth increases (Kang et al., *Frontiers in Marine Science*, 2021, 8: 710891). Diatoms contain a high content of Chls *c* and fucoxanthins associated with FCPs, which enable diatoms to absorb light in the blue-green region, enhancing their ability to survive in deep water. The amount of Chls *c* and Cars in cryptophyte ACPs are far less than that in diatom FCPs, which limits the ability of cryptophytes to absorb blue-green light dominant in deep water and may constrain their growth in deep water. However, this may be compensated somehow by the presence of phycobiliproteins that can absorb blue-green light. In addition, the diadinoxanthin-diatoxanthin (Ddx-Dtx) cycle in diatom FCPs quenches excess energy under strong light conditions in surface water, contributing to nonphotochemical quenching (NPQ) (Goss R and Lepetit B, *Journal of Plant Physiology*, 2015, 172: 13-32). In contrast, the photoprotective Cars, such as alloxanthin, are enriched in cryptophytes, allowing cryptophytes to thrive under high light in shallow water (Mendes et al., *Deep Sea Research Part II: Topical Studies in Oceanography*, 2018, 149: 161-170). Overall, the specific pigment arrangement of ACPIIs, along with the newly identified Chls, leads to the formation of novel EET pathways in cryptophyte PSII–ACPII. The differences in EET pathways of cryptophyte PSII–ACPII and diatom PSII–FCPII due to their distinct LHC structures and arrangements, as well as their distinct pigment compositions, might provide the foundation for cryptophytes and diatoms to thrive in their specific environmental niches.”

18. Line 402 – which chlorophyll atoms were used to calculate the vector of the spectral overlap?

Reply: As shown in Fig. R7 below (also Fig. S15 in the revised manuscript), we selected the porphyrin rings of Chl molecules for the excitation spectra calculation with time-dependent density functional theory (TDDFT). The phytol chain was cut at the site between C1 and C2, and the H atom was adopted to saturate the C1 atom. The excitation energy of each Chl was then used to calculate the spectral overlap using the equations (1.4) and (1.5) as stated in the manuscript. The electronic coupling term was calculated by using the TrEsp (transition charges from electrostatic potentials) method (<https://doi.org/10.1021/jp0615398>) following the equation (1.2). We have added relevant descriptions to the revised manuscript (Page 17 and Fig. S15).

Fig. R7 (Fig. S15). The porphyrin rings of Chl molecules were selected for TDDFT calculation.

19. Figure 5 – please add the subunits marked in the figure and their colors to the legend. In the figure should be “Connector” and not “Conector”.

Reply: We have added relevant descriptions in the legend of Fig. 5, and the typo has been corrected in the revised manuscript.

20. Supplementary table 1 – please add the map vs model CC.

Reply: We have added the map vs model CC values ($CC_{\text{mask}} = 0.83$, $CC_{\text{volume}} = 0.84$ and $CC_{\text{peaks}} = 0.80$) in Table S1.

Reviewers' Comments:

Reviewer #1:

Remarks to the Author:

All my concerns have been addressed; I support the publication of the article.

Reviewer #2:

Remarks to the Author:

The manuscript was satisfactorily revised according to the comments, and is now suitable for publication with one minor correction.

Line 402 – please change LHCII to ACPII.

Responses to Reviewers' Comments

Reviewer #2:

Line 402 – please change LHCII to ACPII.

Reply: Revised.